# Metabolomics in Bariatric and Metabolic Surgery Research and the Potential of Deep Learning in Bridging the Gap

**DOI:** 10.3390/metabo12050458

**Published:** 2022-05-19

**Authors:** Athanasios G. Pantelis

**Affiliations:** Bariatric and Metabolic Surgery Unit, 4th Department of Surgery, Evaggelismos General Hospital of Athens, 106 76 Athens, Greece; ath.pantelis@gmail.com

**Keywords:** metabolic surgery, bariatric surgery, obesity, diabetes mellitus, nonalcoholic steatohepatitis, metabolomics

## Abstract

During the past several years, there has been a shift in terminology from bariatric surgery alone to bariatric and metabolic surgery (BMS). More than a change in name, this signifies a paradigm shift that incorporates the metabolic effects of operations performed for weight loss and the amelioration of related medical problems. Metabolomics is a relatively novel concept in the field of bariatrics, with some consistent changes in metabolite concentrations before and after weight loss. However, the abundance of metabolites is not easy to handle. This is where artificial intelligence, and more specifically deep learning, would aid in revealing hidden relationships and would help the clinician in the decision-making process of patient selection in an individualized way.

## 1. From Bariatric to Metabolic Surgery—A Name Change or a Game-Changer?

Bariatric surgery includes a constellation of surgical procedures that aim at reducing body weight. In current practice, the most commonly performed procedures are laparoscopic sleeve gastrectomy (LSG), laparoscopic Roux-en-Y bypass (RYGB), one-anastomosis gastric bypass (OAGB), single anastomosis stomach–ileal bypass with sleeve gastrectomy (SASI-S), and single anastomosis duodeno-ileal bypass with sleeve gastrectomy (SADI-S) [1,2]. Beyond the primary objective of weight loss, bariatric operations have shown benefits in providing long-term resolution of associated health problems, and most importantly, type 2 diabetes mellitus (T2DM), which is the most prevalent metabolic perturbation worldwide [3,4,5,6]. Consequently, the term “metabolic surgery” has been coined by Professors Henry Buchwald and Richard L. Varco in 1978 and was popularized in the current literature by Professor Francesco Rubino [7,8]. It has been widely adopted by both the bariatric and the endocrinologic community in order to describe the role of weight-loss operations in the armamentarium of antidiabetic interventions offered to people who suffer from diabetes and obesity [9]. This change in terminology reflects the recognition that benefits from operations performed for weight loss are not restricted to reducing body mass index (BMI), but also expand to ameliorating associated medical problems, including type 2 diabetes mellitus (T2DM), hypertension, dyslipidemia, as well as reducing overall mortality [10].

The effect is far from simply mechanistic and includes modulation of neural circuits, alterations of the intestinal microbiome, changes in bile acid excretion, restoration of the intestinal and adipose tissue hormonal milieu (with emphasis on the dynamic balance between incretins and anti-incretins, notably glucagon-like peptide-1 (GLP-1), peptide YY, leptin, ghrelin and glucose-dependent insulinotropic peptide), and recruitment of intestinal glucose transport molecules [7]. Among these mechanisms, the metabolome constitutes a relatively novel, upcoming and promising area of vigorous scientific research in the fields of bariatric and metabolic surgery, as it has been shown in three recent reviews [11,12,13].

## 2. The Metabolome as a Field of Studying the Effects of Bariatric and Metabolic Surgery

Metabolites are a constellation of low-molecular weight molecules (<1 kD) that constitute intermediate or end-products of metabolism. They are classified into amino acid (AA) derivatives (branched-chain amino acids (BCAA—Leu, Ile, Val), aromatic amino acids (AAA—Typ, Phe, Trp), and other amino acids (Gly, Gln, Arg, Orn, Met), lipid derivatives (acylcarnitines, glycerolipids, ketone bodies, phospholipids, and sphingolipids), bile acids (primary (cholic, chenodeoxycholic), secondary (deoxycholic, lithocholic)), microbiota-derived (short-chain fatty acids (SCFA), secondary bile acids, indole compounds, trimethylamine N-oxide (TMAO), tricarboxylic acid cycle (TCA)-related (citrate, pyruvate, succinate), and endocannabinoids (arachidonic acid, 2-arachidonoylglycerol, anandamide) [14]. The entire set of metabolites is collectively named metabolome and belongs to the genetic–phenotypic continuum (genome, transcriptome, proteome), which is also under the dynamic and constant influence of the symbiotic microorganisms of the body (microbiome). Pertinent evidence has increased exponentially over time, and relevant studies may be distinguished into untargeted or targeted ones, according to the metabolomics technique implicated (comprehensive analysis of all metabolites in a sample including unknown ones in the former case vs. measurement of defined and annotated metabolites in the latter). According to a recent review, the top 10 statistically significant metabolomics pathways analyzed in the literature for bariatric and metabolic surgery (BMS) are aminoacyl-tRNA biosynthesis, glycine–serine–threonine metabolism, nitrogen metabolism, phenylalanine metabolism, cysteine–methionine metabolism, TCA cycle, taurine–hypotaurine metabolism, valine–leucine–isoleucine biosynthesis, propanoate metabolism, and nicotinate and nicotinamide metabolism [11]. BCAAs are the most extensively studied compounds in the field of BMS [14].

With regard to BMS, the metabolome has been implicated both as a means to interpret the effects of BMS and as a scaffold of surrogate markers for predicting metabolic and bariatric outcomes on an individualized basis. Regarding the former area of interest, the trends of metabolites following BMS as documented in the literature are as follows—*AA derivatives*: decrease in BCAA (Leu, Ile, Val), AAA (Phe, Tyr, Trpؘ–kynurenine pathway), increase in serotonin, indoxysulfate, indole-3-propionic acid, glycine, and serine; *lipid derivatives*: decrease in free fatty acids (FFA), short-(C3, C5), medium-, and long-chain acylcarnitines, unsaturated and long-chain saturated fatty acids (LCSFA), triglycerides, ceramides, ketone bodies (late postoperatively), increase in medium-chain saturated fatty acids (MCSFA), decanoid acid, phosphatidylcholine, phosphatidylethanolamines, ketone bodies (early postoperatively); *bile acids*: increase in primary and secondary bile acids; *microbiota-related metabolites*: increase in trimethylamine-N-oxide, phenyl sulfate, *p*-cresol; *TCA-related metabolites*: decrease in pyruvate and lactate, increase in citrate, succinate, and malate; *endocannabinoids*: decrease in 2-arachidonoylglycerol, anandamide, arachidonic acid [13,14]. As far as prediction is concerned, relevant studies focus mainly on the relationship of various compounds with T2DM remission [12,15,16,17,18,19], and to a lesser extent, with (suboptimal) weight loss [20,21]. More specifically, successful weight loss was linked to a decrease in AA, a decrease in metabolites of FA metabolism, an increase in 3-hydroxybutyrate, an increase in post-prandial and total glycine amidated-chenodeoxycholic acid, and an increase in post-prandial glycine-amidated hyocholic acid [13]. Similarly, T2DM remission and improved sensitivity to insulin were connected with a decrease in AA (BCAA, AAA, and pyroglutamic acid), a decrease in Trp-derived intestinal microbiota metabolites, a decrease in VLDL, LDL, N-acetyl glycoproteins and unsaturated lipids, an increase in HDL and phosphatidylcholines, a decrease in LCFA (16:0, 18:3 and 17:2), an increase in 3-hydroxybutyrate, a decrease in hippuric acid and 2-hydroxybutyric, and an increase in total bile acids [13].

Ultimately, in a rather breakthrough study, Palau-Rodriguez et al. attempted to correlate the impact of bariatric surgery with the metabolomic profile of bariatric patients, i.e., whether they were metabolically “healthy” (MH) or “unhealthy” (MU) [22]. They found that hydroxy–propionic acids, medium/long-chain hydroxy–fatty acids and bile acid glycuronides were the most discriminative biomarkers of response between MH and MU patients, whereas other metabolites featured various positive or negative correlations with effective weight loss. Furthermore, obesity is considered a state of chronic inflammation [23]. With this regard, C-reactive protein (CRP) could be implemented as a surrogate marker before and after bariatric surgery for evaluating treatment outcomes [24].

As one can understand, thus far, the metabolome is dynamic, not only at discrete time points, but also before and after BMS. The study of the changes of metabolite concentrations as well as their interconversions have recently emerged as a separate field of investigation known as fluxomics. Of importance, experimental animal models have been developed in order to trace incorporation and changes of metabolites after specific types of diet (high carbohydrate vs. ketogenic), by means of isotopic tracing, mass spectrometry, and mathematical analysis [25]. Similarly, fluxomics could be used to elucidate the relationship between incretins/anti-incretins and insulin sensitivity in patients with diabetes before and after metabolic surgery.

## 3. Bariatric and Metabolic Surgery in the Era of Artificial Intelligence

Artificial intelligence (AI) is an umbrella term that describes the process through which a machine (computer) simulates human learning by incorporating (input) a large amount of data (big data), processing them and giving results that lead to conclusions, decisions or adjustments of function (output) [26]. Depending on the degree of human interference in the algorithmic function of artificial intelligence, we have supervised machine learning (unknown input with designated output, predetermined data), unsupervised machine learning (unknown input and output, predetermined data), and deep learning (unknown input and output, data represented in a layered network of neurons). The latter category is the one closest to human learning and has the potential of revealing nonobvious (hidden) relationships between cause and effect (Table 1). There is an increasing body of evidence on AI and relevant publications in healthcare [27].

Our team recently published a scoping review on current evidence regarding the applications of AI in BMS [28]. We identified seven broad categories of subjects: basic science, safety (complications), effectiveness (bariatric outcomes), comorbidities (including T2DM), quality of life, operative characteristics, and cost. Among these, there were only two publications relevant to metabolomics. In the first study, Narath et al. investigated the short- and long-term metabolic changes after bariatric surgery using an untargeted metabolomics approach, with the aid of random forest, an established machine learning (ML) algorithm [29]. In the second study, Candi et al. performed metabolic profiling of visceral adipose tissue from patients living with obesity, with or without metabolic syndrome, who underwent bariatric surgery [30]. Their untargeted metabolomic analysis yielded 481 metabolites, and the results indicated increases in oxidative stress markers (plasmalogens), in addition to changes in glycerolphosphorycholine, glycerolphosphorylethanolamine, glycerolphosphorylserine, ceramides, and sphingolipids. More recently, Perakakis et al. have implemented support vector machine, a supervised learning algorithm, to aid in the diagnosis of nonalcoholic steatohepatitis and fibrosis by noninvasive means of metabolomics [31]. Along the same lines, Castañé et al. investigated the potential of coupling machine learning and lipidomics in order to decipher the metabolic dysfunctions underlying fatty liver disease [32].

Metabolomics datasets are wide, which means that the amount of measurements greatly exceeds the quantity of samples [33]. Nevertheless, ML algorithms depend on large datasets for purposes of training and validation. As such, ML may not be methodologically sound for analyzing metabolomics studies for the time being. Conversely, deep learning (DL) with its simple architecture would be more appropriate. There is a paucity of publications on metabolomics and deep learning in the field of bariatrics, following equally scarce evidence bridging metabolomics and deep learning in general. A metabolomics-specific problem related to this is the fact that metabolites represent highly correlated variables owing to their extensive cross-linking in biochemical processes. Consequently, feature selection is burdensome, and predictive modeling is challenging [33].

## 4. Deep Learning and Metabolomics—Too Hard to Handle or the Coming of an Era?

In the past, there have been efforts to apply DL in metabolomics data acquisition and processing, stratification of metabolic phenotypes, integration of metabolomics into multi-omics studies, prediction of metabolic pathways, and genome-wide metabolic modeling [34]. These efforts have met several methodological predicaments, such as high computational cost, suboptimal training and internal validation, lack of external validation, noncalculation of isotopic peaks (spectrometry and its variations are pivotal in the quantification of metabolites), overfitting in case of applying DL to data with low sample size, reduced predictive ability upon application to class-imbalanced datasets, poor applicability of experimental models to human metabolism, etc. [34]. Date and Kikuchi tried to overcome some of these barriers by applying a mean decrease accuracy (MDA) calculation in a deep neural network (DNN) [35]. Here is an attempt to systematize current limitations of combining DL with metabolomics and through these make BMS-specific suggestions with clinical orientation. (1)**Lack of human-centric data availability**: Bariatric patients constitute a surgical population that is submitted to vigorous follow-up. Most importantly, there are two large international bariatric databases (IFSO^®^ Registry and MBSAQIP^®^ Database) that are regularly updated. Starting to integrate metabolic profiles of patients before and at discrete follow-up visits after BMS apart from demographic, clinical, routine laboratory, and nutrition data would serve as a scaffold for a population-wide metabolomics database.(2)**Dimensionality and overfitting**: This limitation is generated by the asymmetrical distribution of low numbers of samples and too many measured features in the context of metabolomics (high-dimension low-sample size data; HDLSS). Again, the key to this could be found in the large populations of existing bariatric databases, provided they start to integrate data on metabolites.(3)**Lack of metabolomics-specific DL features**: This problem is non-BMS specific, but it is rather ubiquitous for metabolomics. In contradistinction to genomics and proteomics, metabolomics still lacks well-defined problem statements and methods. Relevant studies in the aforementioned fields employ strategies that convert DNA/RNA sequences and protein structures (alpha-helix, beta-strand, loop region) into encoded representations that are suitable for convoluted neural network (CNN) applications [33]. More advanced is the conversion of nonimage data to image-like data suitable for processing with CNN [36].(4)**Challenging model validation**: Given that metabolomics datasets are considered HDLSS, one measure to overcome the current lack of large populations would be to apply nonconventional validation methods. For example, a widely adopted validation method, *k*-fold cross validation, leads to biased performance with small sample sizes. On the contrary, nested cross-validation has shown stable performance regardless of the sample size [37]. Metabolomics studies on MBS patients with limited sample sizes could benefit from this approach.

Applying DL to the metabolome with respect to BMS would be a revolutionary endeavor. A DL algorithm with its layered architecture would be able to process the abundance of metabolites and reveal hidden “outside the box”, nonlinear relationships between them and bariatric and/or metabolic outcomes. These results could be further used to predict each individual patient’s postoperative course and response to each bariatric surgery according to their metabolomic setup. In conjunction with other omics, this process would become more accurate and individualized. With respect to the disciplines mentioned earlier, DL could provide specific benefits: (1)**Basic science**: Existing studies on retrieving metabolites and their pathophysiological role could benefit from a layered structure of analysis after applying the strategies mentioned earlier. A good relevant example is the study of Date and Kikuchi [35]: their experimental model was yellowfin goby (*Acanthogobius flavimanus*), their objective was metabolic characterization depending on geographical distribution, the sample type was muscle tissue, they studied two sets of metabolites (water-soluble components, *n* = 170; methanol-soluble components, *n* = 1022), and the utilized DL algorithm was a deep neural network analytical approach that yielded better classification accuracy as compared to ML algorithms. Analogous experiments could be implemented to existing animal models of BMS [38,39] with the purpose of metabolomic characterization before and after initial operation and monitoring of BMS effectiveness at a second level.(2)**Safety (complications):** DL algorithms could be implemented to map the metabolome of BMS candidates and to correlate specific metabolotypes with certain adverse outcomes, such as defective wound healing that may lead to a leak or to susceptibility to the formation of adhesions.(3)**Effectiveness (bariatric outcomes):** Weight loss is the main objective of the vast majority of BMS. Consequently, being able to predict in advance which patient is going to benefit from which operation based on their metabolomic synthesis would be of utmost research and clinical interest. DL could be implemented in order to yield data from existing populations and through hidden layers that reveal favorable and avoidable metabolomic setups.(4)**Associated medical problems (including T2DM):** Existing evidence focuses on the metabolic aspects of metabolomics on patients who undergo BMS [22,29]. The advent of DL, following the implementation of strategies to overcome limitations, could contribute to a more widespread performance of such studies.

All bariatric patients who are prepared for BMS are submitted to extensive laboratory workup, which includes complete blood count, coagulation studies and biochemical studies, both for ensuring suitability for surgery and for obtaining baseline values on sugar metabolism, lipid profile, and micronutrient sufficiency. At the immediate postoperative period, most patients undergo blood tests in the context of monitoring the impact of the operation itself and for promptly diagnosing potential adverse outcomes (hemorrhage, acute kidney injury, liver dysfunction, etc.). More frequently than not, virtually all accredited bariatric programs integrate mandatory blood testing at regular intervals (i.e., 3, 6, 12, 24 months postoperatively) as part of their follow-up scheme, for the purpose of monitoring the metabolic and nutritional impact of the bariatric operation. Consequently, blood is the most readily available biological specimen. It is understandable that blood may not be the most appropriate medium for assessing specific metabolites, in part because of the interference with other plasma proteins and circulating mediators [40]. Even so, blood requires careful handling and pre-analytical processing in order to be amenable to metabolomics analysis [41]. It has been postulated that BMS provides a unique opportunity to obtain both tissue biopsies (from the liver, adipose tissue, jejunum, or from the resected stomach, in the case of sleeve gastrectomy) and portal vein blood samples [42]. Regardless of their theoretical superiority, such endeavors entail a certain learning curve, lead to extension of the operative time and bear a small but nonnegligible complication burden. As such, for the time being, the most appropriate biological sample considered for human metabolomics studies in MBS should be blood. Microbiota-related metabolomics studies could also include feces, which is relatively easy to collect and process. Figure 1 summarizes the proposed workflow algorithm for metabolomics studies in BMS.

Regarding BMS outcomes, the major stake of clinical prediction is prognosis rather than diagnosis, and this is relevant to both diabetes remission and weight loss or regain. The ultimate purpose for the clinician is to be able to determine preoperatively which patient will benefit from a specific BMS procedure and who will not but, on the contrary, be exposed only to the perils of an operation. This way, DL-based metabolomics studies will have a role in the preoperative decision-making process, in what is assumed as a typical classification problem (responders–nonresponders). Conversely, prediction in safety studies is relevant to both prognosis and diagnosis, but the clinical benefit of the latter remains to be shown, given that more practical approaches are readily available.

The two major drawbacks of deep learning are the high demand of computational power and the fact that the relationships simulated by neural networks may be less self-explanatory than the determined algorithms of conventional machine learning. These problems are ubiquitous across data analyses. From a clinical point of view, only head-to-head comparisons between the two types of methodologies may reveal the advantages of one or the other. Again, the study of Date and Kikuchi give such an example, where a DNN-based DL algorithm that was considered inappropriate for classification and regression modeling as compared to its ML counterparts was rendered appropriate after implementation of importance estimation for each variable using MDA calculation [35].

The significant impact that DL is beginning to make on metabolomics data processing and analysis paves the way for the future. Most importantly, it underlines the importance of integrating data analysts into the multidisciplinary team which is dedicated to the care of patients living with obesity. Close collaboration between clinicians and data scientists could expedite the adoption of DL into daily practice for the benefit our patients.

## Figures and Tables

**Figure 1 metabolites-12-00458-f001:**
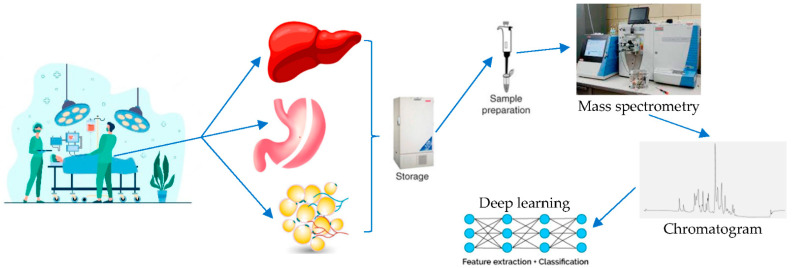
A schematic model of how tissue retrieved during bariatric metabolic surgery can be used for metabolomic analysis, and consequently, the resulting data undergo feature extraction and classification with deep learning techniques. As a first step, metabolomics analyses on tissue samples could be compared against metabolomic analyses on body fluids before operation (serum, urine, feces). At a following stage, fluctuations in metabolite concentrations in body fluids could be measured at standardized intervals after the bariatric operation and compared to both preoperative body fluid values, as well as tissue sample values. At a final stage, changes in metabolite concentrations could be correlated with clinical data that are routinely collected before and after bariatric surgery, such as BMI, blood glucose, HbA1c, HDL, LDL, total cholesterol, and triglycerides, etc.

**Table 1 metabolites-12-00458-t001:** The main types of artificial intelligence with examples of statistical approaches for each one.

Type of AI Algorithm	Purpose	Examples
Supervised machine learning	Classification (categorical output, i.e., obese, not obese, T2DM remission-nonremission) or Regression (continuous output, i.e., weight, BMI, HbA1c level).	Decision trees, random forest, knn, logistic regression
Unsupervised machine learning	Clustering (inherent grouping in data, i.e., grouping responders of bariatric surgery based on their metabolomic setup) or Association (discovering the rules that describe large portions of data).	K-means for clustering, a priori algorithms
Deep learning	Input and output are connected in layers with relationships that resemble neural networks in the nervous system. These relationships are usually “hidden”.	Convolutional neural networks, artificial neural networks, Bayesian networks.

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
