# Peer review of "Metabolomics in Bariatric and Metabolic Surgery Research and the Potential of Deep Learning in Bridging the Gap"

_metabolites, 2022, doi:10.3390/metabo12050458_

Round 1

Reviewer 1 Report

People with with extreme weight groups are at least partially characterized by systemic inflammation. The available literature shows that the lowest levels of CRP are found in people with normal body mass, and the relationship between BMI and proinflammatory factors takes the form of the letter U (1). Therefore, it is worth noting the need for CRP level determination prior to qualification for bariatric treatment, as there are data suggesting that persistent inflammation may be associated with poorer treatment outcomes (1).

The manuscript is well written and a stimulus for the readership.

Please add the following reference:

1. Doi: 10.2217/bmm-2018-0101

Reviewer 2 Report

I read with interest the review by Pantelis on metabolomics in bariatric and metabolic surgery research and the potential of deep learning in bridging the gap.

I think that this is an interesting manuscript in this new era of metabolomics as potential approach for discovering new diagnostic and prognostic biomarkers in the setting of bariatric surgery. Weight loss impacts on metabolism and several metabolic profile can be associated with the wide spectrum of post-operative conditions. Moreover, machine learning is a foundamental approach for big data analysis necessary to better understand potential physiopathological mechanisms.

Minor comments

  • I suggest to include a paragraph on fluxomic. The study of metabolic flexibility by glucose and lipids fluxes is important to find alterations associated with insulin resistance, a condition that frequently characterizes obese patients. For example, It is known that bariatric surgery improves glucose homeostasis in patients with type 2 diabetes even before any significant weight loss is achieved. Moreover,  fluxomic can be used to elucidate the relationship between hormones concentrations and insulin sensitivity in obese subjects without diabetes and to assess the relation between these hormones (such as the incretine GLP-1) and the improvements in glucose metabolism following bariatric surgery.
  • I suggest to add a Table describing the main statistical approaches by machine learning with some examples because this is another important part of the manuscript.

Reviewer 3 Report

The opinion article presented by Athanasios G. Pantelis entitled Metabolomics in Bariatric and Metabolic Surgery Research and the Potential of Deep Learning in Bridging the Gap aims to highlight the relevance of using AI in the analysis of the metabolome study of Bariatric and Metabolic Surgery patients.
In my opinion, for the scope of the journal it would be necessary to explain in more detail Bariatric and Metabolic Surgery since the specialty of the journal is Deep Learning for Metabolomics the relevance might be misunderstood. More interconnection between the topics should be done.
In my opinion AI will be of relevant application in all data involving clinical data and metabolomic data. The opinion article does not explain in a schematic and detailed way what types of data will be used and how it will be handled. Could a little more detail and an additional schematic help?
From what is said in the article there are still few articles/results on this topic. In my opinion an opinion article should be more detailed/specific to help who wants to work in the area.
The big advantage of metabolomics is the fact that it can be applied in a nonvasive way. The author has some doubts regarding the use of blood (plasma or serum) and encourages the use of biopsies. Can a joint analysis and translation of information be done? How will it be integrated with other demographic and clinical data?
The sentence "Among these mechanisms, the metabolome constitutes a relatively novel, upcoming and promising area of vigorous scientific re- search." In my opinion it should be explained in more detail.
The sentence "The entire set of metabolites is collectively named metabolome and belongs to the genetic-phenotypic continuum (genome, transcriptome, proteome)." is not completely correct. The microbiome also influences the metabolome. 
The document has some misplaced phrases, for example: lines 62-63 and lines 90-96. A more careful reading will be necessary.

Round 2

Reviewer 3 Report

Author have addressed my major concerns. I think this study is acceptable now.